# Second-Trimester Constituents of the Metabolic Syndrome and Pregnancy Outcome: An Observational Cohort Study

**DOI:** 10.3390/nu14142933

**Published:** 2022-07-18

**Authors:** Jonas Ellerbrock, Esmee Hubers, Chahinda Ghossein-Doha, Veronique Schiffer, Robert-Jan Alers, Laura Jorissen, Jolijn van Neer, Maartje Zelis, Emma Janssen, Sabine Landewé-Cleuren, Annemie van Haarlem, Boris Kramer, Marc Spaanderman

**Affiliations:** 1School for Oncology and Developmental Biology (GROW), Maastricht University, Universiteitssingel 50, 6229 ER Maastricht, The Netherlands; chahinda.ghossein@mumc.nl (C.G.-D.); veronique.schiffer@maastrichtuniversity.nl (V.S.); robertjan.alers@mumc.nl (R.-J.A.); laura.jorissen@mumc.nl (L.J.); emma.janssen@mumc.nl (E.J.); marc.spaanderman@mumc.nl (M.S.); 2Department of Obstetrics and Gynecology, Zuyderland Medical Center, H. Dunantstraat 5, 6419 PC Heerlen, The Netherlands; m.zelis@zuyderland.nl; 3Department of Obstetrics and Gynecology, Radboud University Medical Center, Geert Grooteplein Zuid 10, 6525 GA Nijmegen, The Netherlands; esmeehubers@gmail.com; 4Department of Cardiology, Maastricht University Medical Center, P. Debyelaan 29, 6229 HX Maastricht, The Netherlands; 5Department of Obstetrics and Gynecology, Maastricht University Medical Center, P. Debyelaan 29, 6229 HX Maastricht, The Netherlands; jolijn.van.neer@mumc.nl; 6Department of Internal Medicine, Maastricht University Medical Center, P. Debyelaan 29, 6229 HX Maastricht, The Netherlands; s.landewe@mumc.nl (S.L.-C.); a.van.haarlem@mumc.nl (A.v.H.); 7Department of Pediatrics, Maastricht University Medical Center, P. Debyelaan 29, 6229 HX Maastricht, The Netherlands; b.kramer@mumc.nl

**Keywords:** gestational diabetes, dyslipidemia, hypertension, obesity, preeclampsia, preterm birth, metabolic syndrome

## Abstract

Background: Gestational diabetes mellitus (GDM) increases the risk of type 2 diabetes mellitus and cardiovascular disease (CVD) in women in later life. In the general population, metabolic syndrome (MetS) shows identical associations. The aim of this study was to evaluate the association between GDM, constituents of MetS and pregnancy outcomes. Methods: Of 2041 pregnant women undergoing an oral glucose tolerance test (OGTT) between 22 and 30 weeks of gestation, data were collected to evaluate the constituents of MetS. Odds ratios (ORs) were calculated to determine the associations between MetS and pregnancy outcomes. Results: GDM and obesity did not affect the risk of fetal growth abnormalities (SGA/LGA), preterm birth or preeclampsia (PE). Hypertension significantly increased the risk of SGA (OR—1.59), PE (OR—3.14), and preterm birth <37 weeks (OR—2.17) and <34 weeks (OR—2.96) and reduced the occurrence of LGA (OR—0.46). Dyslipidemia increased the risk of PE (OR—2.25), while proteinuria increased the risk of PE (OR—12.64) and preterm birth (OR—4.72). Having ≥2 constituents increased the risk of PE and preterm birth. Conclusions: Constituents of metabolic syndrome, rather than treating impaired glucose handling, increased the risk of preeclampsia, altered fetal growth and preterm birth. Obesity was not related to adverse outcomes.

## 1. Introduction

Gestational diabetes mellitus (GDM) is one of the most prevalent medical complications in pregnancy that affects, depending on the population studied, up to 25% of all pregnancies [1]. GDM is defined as any degree of glucose intolerance with onset or first recognition during the second half of pregnancy. If not detected or properly treated, GDM increases the risk of maternal and neonatal complications, including gestational hypertensive disease, fetal growth disorders and preterm delivery [2]. The timely detection and treatment of GDM reduces the rates of macrosomia and macrosomia-related complications, providing similar neonatal outcomes as to nondiabetic pregnancies [2].

Overweight and obesity are currently the dominant risk factors for GDM [3]. Maternal obesity is also associated with an increased risk of gestational hypertensive disease and preterm birth [4]. On the one hand, increased maternal weight is associated with dyslipidemia, impaired glucose handling, raised maternal glucose levels and fetal glucose availability, giving rise to accelerated fetal growth [4]. On the other hand, increased body weight is associated with high blood pressure, dyslipidemia and proteinuria, all factors that predispose one to gestational hypertensive disease, preterm birth and attenuated fetal growth [5]. Therefore, these divergent cardiometabolic and cardiovascular risk factors associated with increased body weight and consistent with metabolic syndrome may underlie the various gestational clinical phenotypes observed [6,7]. 

Metabolic syndrome (MetS) results in impaired glucose handling, raised blood pressure, dyslipidemia, obesity and proteinuria [8]. As during pregnancy, physiological changes affect glucose tolerance, blood pressure, lipids, body mass and renal function, the nonpregnant defined cut-off values of the constituents of metabolic syndrome should be adjusted to the expected cardiovascular and cardiometabolic changes in pregnancy [9,10,11]. The predictive strength of these different constituents determined in the second trimester of pregnancy with regard to fetal growth, maternal hypertensive complications and preterm birth remains to be elucidated. 

The aim of this study was to unravel the relation of different constituents of metabolic syndrome and impaired gestational glucose tolerance with subsequent maternal and neonatal outcomes.

## 2. Materials and Methods

### 2.1. Study Design and Population

This is a prospective cohort study on pregnant women who underwent routine oral glucose tolerance testing (OGTT) in the Maastricht University Medical Center (MUMC+) according to the Dutch national guidelines in the second trimester of their pregnancy. Women were excluded when they met the following criteria: hospitalization for early-onset preeclampsia before OGTT, location of delivery other than MUMC+, gestational age <22 or >30 weeks, incomplete diagnostic information because of vomiting during the test or missing medical records, and/or multiple pregnancies (twins or triplets). In the case of multiple testing, only data of the screening assessment were used in this study.

### 2.2. Anthropometric Data

Ethnicity and pre-pregnancy weight were self-reported. Height was measured at the time of the OGTT. Pre-pregnancy weight was used to calculate pre-pregnancy body mass index (pre-BMI), while weight at the assessment day was used to calculate pregnant BMI. Blood pressure was measured in a quiet environment in a sitting position, using an oscillometric method (Carescape V100, GE Healthcare, Milwaukee, WI, USA) for 30 min at 3 min intervals. Median value was used for analysis.

### 2.3. Biochemical Analysis

All participating women underwent OGTT screening between 22 and 30 weeks of pregnancy. Maternal venous blood samples were drawn in the morning after an overnight fast to assess their metabolic profiles. In addition, while still fasting, a urine sample was collected.

Venous blood sampling was followed by ingestion of a 75 g glucose drink (82.5 g Dextrose monohydrate by Fagron, Rotterdam, The Netherlands). Additional blood samples were collected for glucose level determination 1 and 2 h post load.

Fasting plasma glucose (FPG), glycolyzed hemoglobin (HbA1c), total cholesterol, high-density lipoprotein (HDL), triglycerides (TG) and proteinuria (protein-creatinine ratio) were measured using an autoanalyzer (Cobas 8000 Roche, Basel, Switzerland). Fasting insulin was analyzed using an immune-assay (Imulite XPi, Siemens Healthineers, Erlangen, Germany).

### 2.4. Diagnosing the Metabolic Syndrome

To diagnose metabolic syndrome in pregnancy, the World Health Organization (WHO) criteria were adapted to pregnancy. The WHO defines metabolic syndrome as the presence of abnormal glucose handling plus two of the following: obesity, dyslipidemia, hypertension and/or proteinuria [12]. Published reference values in pregnancy were used to adapt the values to pregnancy and were considered abnormal when 2SD above (blood pressure, triglycerides) or below (HDL) the gestational age adjusted reference range [13,14,15]. 

Abnormal glucose handling (insulin resistance) was measured by OGTT. Abnormal OGTT was diagnosed when fasting glucose ≥5.2 mmol/L and/or 1 h plasma glucose ≥10.0 mmol/L and/or 2 h plasma glucose ≥8.5 mmol/L [16].

We determined hypertension to be when systolic blood pressure ≥130 mmHg and/or diastolic blood pressure ≥80 mmHg or when taking antihypertensive drugs (Labetalol, Nifedipine, Methyldopa, Metoprolol) [13]. Obesity was defined by a pre-pregnancy BMI of ≥30 kg/m^2^. 

Dyslipidemia was diagnosed when HDL cholesterol <1.1 mmol/L and/or triglycerides ≥3.5 mmol/L [14]. Proteinuria was defined by a protein to creatinine ratio of ≥30 g/mol creatinine [15].

### 2.5. Treatment during Pregnancy

Women diagnosed with GDM were first treated with a carbohydrate-restricted diet. If insufficient glucose control was reached during follow-up, women were treated with metformin, insulin or both until adequate glucose control was achieved. We considered glucose control adequate when fetal growth and/or fetal abdominal circumference did not change more than 40 centiles from that observed at 18–20 weeks gestational age ultrasound measurement, when amniotic fluid was not increased above the 95th centile and when maternal blood glucose levels were within the target range (fasting capillary blood glucose ≥ 5.3 mmol/L and 1 h post-prandial ≥ 7.8 mmol/L).

### 2.6. Adverse Pregnancy Outcomes

Pregnancy outcomes were collected from patient files, including birthweight and centile, occurrence of preeclampsia and preterm birth. These outcomes were only available if the woman had given birth in the hospital. A neonate was considered small for gestational age (SGA) when birth weight was below the 10th centile and large for gestational age (LGA) when birth weight was over the 90th centile, using age- and sex-specific national growth charts [17]. Preeclampsia, HELLP syndrome and eclampsia were defined according to the ‘National High Blood Pressure Education Program Working Group Report on High Blood Pressure in Pregnancy’ [18]. Preterm birth was viewed as delivery prior to <37 weeks and prior to <34 weeks of gestation.

### 2.7. Statistical Analysis

Statistical analyses were conducted using SPSS (Version 24.0.1.0, IBM Corp, Armonk, NY, USA). Quantitative values were expressed as mean with standard deviation (SD) when normally distributed and median and IQR when not normally distributed. Differences were tested parametrically and nonparametrically, whenever applicable. ANOVA and chi-square tests or Fisher’s exact test along with odds ratio (OR), whenever applicable, were used to evaluate univariate differences between continuous and categorical variables, respectively. A complete case analysis using logistic regression was performed to evaluate multivariate differences by estimating ORs adjusted (aORs) for (other) constituents of metabolic syndrome, age, Northern European ancestry and gestational age at time of OGTT. A Mantel–Haenszel test for trend analysis was used to test trends between categorical variables. A *p*-value < 0.05 was considered statistically significant.

## 3. Results

### 3.1. Study Population

In total, 3245 OGTTs were performed between January 2014 and December 2019. After selection, 2041 participants were included in the analysis (Appendix A). Complete case multivariable analysis could be performed for 1984 women (97.2%).

The baseline characteristics are presented in Table 1. Women were on average 31.6 ± 5.0 years of age with a pre-pregnancy BMI of 27.6 ± 7.8. The primary ethnicity was Northern European (79.6%).

Of the total studied population, 28.2% were diagnosed with GDM. Moreover, 30.4% were obese prior to pregnancy and 60.9% were at least overweight. Hypertension was present in 17.5% of participants, dyslipidemia in 5.7% and proteinuria occurred in 1.3% of cases. Metabolic syndrome during the second trimester of pregnancy was present in 4.3% of the studied population. In the entire screened population, 4.8% developed preeclampsia, 8.3% delivered prior to the 37th week of gestation, and 2.7% before the 34th week. Of all neonates, 10.3% were diagnosed as LGA, while 9.8% were SGA. Intra-uterine fetal demise occurred in 0.2%.

Maternal age, height, pre-pregnancy weight and pre-pregnancy BMI, blood pressure and the prevalence of dyslipidemia, overweight and obesity were significantly higher in women with GDM as compared to without GDM. When diagnosed and treated for GDM, neither gestational maternal hypertensive sequelae nor offspring birthweight, centile or prematurity occurred differently in both groups. In addition, no difference in operative or instrumental delivery, shoulder dystocia, placental adherence problems or peri-partum blood loss was observed between both groups.

### 3.2. The Effect of the Number of Constituents

Table 2 shows the occurrence of adverse pregnancy outcomes related to the number of constituents of the metabolic syndrome present. In both women with and without GDM, statistically, fetal growth remains unaffected at increasing concurrent constituents of metabolic syndrome. In contrast, irrespective of the presence or absence of GDM, preeclampsia and preterm birth (<37 weeks and <34 weeks) occurred more with an increasing presence of risk constituents.

To explore the effect of GDM itself and concurrent constituents hypertension, obesity, dyslipidemia and proteinuria on the outcomes LGA, SGA, PE and preterm birth (<37 and <34 weeks), we calculated (adjusted)ORs (95% CI) in the absence or presence of GDM (Table 3, Appendix A).

#### 3.2.1. GDM

GDM, when detected and treated, did not significantly affect the risk of LGA, SGA, PE, preterm birth <37 and <34 weeks in the total group. In women with obesity, the presence of GDM decreased the occurrence of SGA offspring (aOR—0.48 (CI—0.25–0.94)). This decrease in SGA was also seen in women without proteinuria (OR—0.68 (CI—0.48–0.98)), although this effect was not statistically significant after correction for metabolic syndrome constituents, age, Northern European ancestry and gestational age.

#### 3.2.2. Hypertension

Hypertension decreased the risk of LGA in the total group (aOR—0.46 (CI—0.28–0.77)), and increased the risk for SGA (aOR—1.59 (CI—1.09–2.31)), PE (aOR—3.14 (CI—1.94–5.10)), preterm birth <37 weeks (aOR—2.17 (CI—1.46–3.23)) and <34 weeks (aOR—2.96 (CI—1.59–5.53)), largely irrespective of the presence of coexisting GDM.

#### 3.2.3. Obesity

In the total population studied, obesity itself did not significantly affect any of the pregnancy outcomes evaluated. Only amongst women with GDM did the presence of obesity decrease the risk of SGA (OR—0.48 (CI—0.25–0.92)), which, however, did not remain statistically significant after adjustment (aOR—0.52 (CI—0.26–1.07)).

#### 3.2.4. Dyslipidemia

Dyslipidemia did not significantly affect the risk of altered fetal growth but affected the risk of PE (aOR—2.25 (CI—1.09–4.73)) and preterm delivery; the latter effect was not significant after adjustment.

#### 3.2.5. Proteinuria

Proteinuria increased the univariable risk of subsequent SGA in the total group (OR—2.79 (CI—1.11–7.03)), although not significant after adjustment (aOR—1.84 (CI—0.60–5.70)).

Proteinuria strongly independently increased the risk of PE in the total group (aOR—12.64 (CI—4.37–36.54)), an association observed in both women without GDM (aOR—4.33 (CI—0.96–19.63)) and the women with GDM (aOR—56.85 (9.02–358.46)).

Largely irrespective of glucose metabolic disorders, proteinuria increased the risk of preterm birth <37 weeks (aOR 4.72 (CI—1.67–13.35)) and preterm birth <34 weeks (aOR—5.16 (CI—1.33–20.08)).

#### 3.2.6. Two or More Constituents

Having ≥2 constituents of metabolic syndrome did not significantly affect the risk of LGA or SGA in the total group, but increased the risk of PE in the total group (OR—3.07 (CI—1.86–5.07)), irrespective of GDM. Having ≥ 2 constituents also increased the risk of preterm birth both <37 and <34 weeks in the total group (OR—2.01 (CI—1.29–3.13) and OR—3.20 (CI—1.68–6.10), respectively), a relation that occurs regardless of the presence of GDM.

## 4. Discussion

### 4.1. Main Findings

We observed that, when detected and treated, GDM hardly affected clinical outcomes. In contrast, high blood pressure, dyslipidemia and proteinuria profoundly negatively affected maternal and offspring wellbeing. Although the effect of multiple concurrent constituents on fetal growth did not reach statistical significance, the negative impact on maternal health (preeclampsia) and preterm birth was significant. Obesity itself did not affect any of the evaluated clinical outcomes.

### 4.2. Strengths and Limitations

This is one of the first clinical observational cohort studies that integrally investigated the contribution of cardiovascular and metabolic risk factors during routine second-trimester OGTT. The studied sample size enabled us to unravel the divergent effects of these risk factors on maternal and offspring gestational outcomes. Herewith, it at least partly explains the different clinical translates reported in women diagnosed with GDM.

There are also several limitations that need to be addressed. First, the study population consisted mainly of women of Northern European ancestry, which could affect generalizability. On the one hand, others observed that the prevalence of metabolic syndrome varies substantially between ethnic groups, even after adjusting for BMI, age, socioeconomic status and other variables [19]. On the other hand, underlying risk factors rather than BMI itself affected outcome. Therefore, despite possible genetic differences and in line with our ancestry-adjusted observations, the composition and presence of risk factors rather than solely ethnicity may be responsible for the outcome [20]. 

Second, all patients with GDM underwent tight management with dietary interventions and, if necessary, medicament treatment. These interventions most likely have normalized fetal growth, but may also have affected weight gain, lipid composition, blood pressure development and the prevalence of preterm birth [21]. Undetected and untreated GDM may relate differently to all described outcomes. Additionally, correlates of most additional risk variables behave comparably between those with and without GDM.

### 4.3. Interpretation

#### 4.3.1. GDM

GDM originates from placenta-induced increased insulin resistance raising glucose availability transported across the placenta. Consequently, fortified fetal glucose storage and the stimulated endogenous production of insulin-like growth factor (IGF-1), both contributing to overgrowth, raise and lower the prevalence of LGA and SGA, respectively [3]. 

It has been hypothesized that insulin resistance contributes to the pathophysiology of PE. Many PE risk factors are also associated with insulin resistance, including chronic hypertension, advanced maternal age, GDM, DM and obesity, at least partially explaining the observed association [22]. Previous studies have suggested that insulin resistance is an independent predictor of PE after adjustment for concurrent risk factors [23]. Additionally, the risk of preterm birth increases with increasing levels of glycemia [24]. High glucose levels negatively affect vascular function and may mimic the early phase towards the development of diabetic vasculopathy. Diabetic vasculopathy is associated with PE, growth restriction and mostly iatrogenic preterm birth [25]. Prior studies show that GDM raises the risk of LGA (OR—1.8 (CI—1.7–1.8)), PE (OR—1.7 (CI—1.6–1.7)) and preterm birth (OR—1.3 (CI—1.3–1.4)) [26]. In line with these findings, we expected GDM to increase the risk of several adverse pregnancy outcomes. However, we did not find any significant change with regard to pregnancy outcomes in detected and treated GDM; neither LGA, SGA or PE, nor preterm birth was statistically significant different between women without or tightly managed GDM. Apparently, for these outcomes, detection and treatment lowers the risk of these adverse outcomes as observed during studied conditions providing similar neonatal outcomes as to nondiabetic pregnancies [2].

#### 4.3.2. Hypertension

Hypertension is associated with circulatory maladjustments to pregnancy, often paralleled by suboptimal placentation and increased endothelial shear, all factors that independently contribute to remote maternal hypertensive sequelae, attenuated fetal growth and preterm birth [27]. Hypertension is reported to raise the risk of SGA (OR—2.06 (CI—1.79–2.39)), PE (OR—5.76 (CI—4.93–6.73)) and iatrogenic preterm birth (<37 weeks) (OR—3.73 (CI—3.07–4.53)) and lowers the prevalence of LGA (OR—0.65 (CI—0.53–0.76)) [28]. Our findings are consistent with these observations, but importantly, the effects are independent of and unaffected by concurrent glucose disturbances.

#### 4.3.3. Obesity

Maternal obesity is associated with several factors affecting gestational outcomes with an increased activity of molecular IGF-1 signaling pathways and inflammatory pathways, increased sympathetic tone, attenuated insulin signaling, altered central hemodynamic functions, and increased cardiometabolic risk factors, all independently capable of affecting placental and endothelial wellbeing [29,30].

Obesity is reported to increase the risk of PE, LGA, and preterm birth [31]. Despite these associations, we observed only a halved prevalence of SGA in women with GDM, an effect that primarily seems to rely on attenuated glucose handling as the prevalence of SGA is only seen in obese women with GDM compared to those with normal glucose tolerance. For all other adverse pregnancy outcomes, obesity did not show altered risks. Theoretically, it may be that the treatment of concurrent GDM reduced these risks. Dreisbach et al. showed that diet interventions in obesity change the intestinal microbiota, which in time can influence molecular signaling and inflammation in pregnant women and their offspring, possibly overshadowing the negative effects of obesity [32]. From earlier studies, it is known that certain diet interventions can favorably alter metabolic outcome in obesity, even without weight loss [33]. This could explain obesity not being the negative influencing factor with regard to pregnancy outcome but its metabolic effects.

Additionally, although controversial, obesity is not synonymous with metabolic syndrome. Some consider obese individuals as metabolically healthy (MHO) when there are no other symptoms of metabolic syndrome [34]. MHO is more common in women than in men, in younger than in older adults and in people of European ancestry [35]. Our study population consists of young women, predominantly of Northern European ancestry. Therefore, either because of a favorable profile or as a consequence of the dietary intervention, these women were, or became, metabolically healthy.

#### 4.3.4. Dyslipidemia

Dyslipidemia has divergent effects on pregnancy. From a maternal point of view, low HDL or elevated triglycerides relates to endothelial oxidative stress and inflammation [36], whereas from a fetal point of view these lipid conditions raise nonesterified fatty acid availability for placental transfer, an important condition for fetal growth, for which triglycerides are the main source [37]. Various studies found these effects to increase the risk of PE (OR—1.5 (CI—1.16–1.93)), LGA (OR—1.13 (CI—1.02–1.26)) and preterm birth (OR—1.49 (CI—1.39–1.59)) and decrease the risk of SGA (OR—0.63 (CI—0.40–0.99)) [38,39]. We also observed a more than doubled risk of PE in the total group and an almost doubled risk of preterm birth, but, despite the anticipated direction on offspring weight, this did not reach statistical significance.

#### 4.3.5. Proteinuria

Towards the development of placental syndromes, concurrent inflammation, endothelial cell dysfunction, oxidative and shear stress-induced placental damage increases capillary leakage, the loss of circulatory volume, global maternal circulatory dysfunction, ultimately leading to hypertensive sequelae, attenuated fetal growth and (iatrogenic) preterm birth [40]. Proteinuria detected by random sampling relates to later development of PE, preterm labor and SGA [41]. In chronic hypertensive pharmacological-treated women, concurrent proteinuria was reported to increase the risk of later PE (79% vs. 49%), preterm birth (48% vs. 26%) and SGA (41% vs. 22%) compared to chronic hypertensive women without proteinuria [42]. In line with this, irrespective of blood pressure, we observed a substantially increased risk of SGA (OR—2.79 (CI—1.11–7.03)), PE (OR—13.68 (CI—6.03–31.03)), preterm birth <37 weeks (OR—7.36 (CI—3.29–16.50)) and <34 weeks (OR—9.52 (CI—3.45–26.32)) in women with as compared to without second trimester proteinuria.

Comparable to earlier studies, we observed that the risk of adverse pregnancy outcomes increases when other cardiovascular and cardiometabolic risk factors were simultaneously present [6]. Irrespective of normal or abnormal glucose handling, we did not observe a statistically significant relation of the concurrent presence of these factors on SGA or LGA; two or more simultaneously present constituents of metabolic syndrome doubled the risk of preterm birth <37 weeks, tripled the risk of preterm birth <34 weeks and fourfold increased the risk on PE.

## 5. Conclusions

Besides attenuated glucose handling, concurrent constituents of metabolic syndrome, hypertension, dyslipidemia and proteinuria, all cumulatively increase the risk of the later development of preeclampsia, altered fetal growth and preterm birth. Obesity itself did not show any relation to these adverse outcomes. Nonetheless, with increasing body weight and its prevalence of underlying metabolic syndrome, women may benefit from more targeted interventions normalizing these possible concurrent risk factors to improve pregnancy outcomes for both mother and child.

## Figures and Tables

**Table 1 nutrients-14-02933-t001:** Baseline characteristics and pregnancy outcome in women subjected to routine screening second-trimester oral glucose tolerance test either diagnosed with or without gestational diabetes mellitus.

	Total	Gestational Diabetes Mellitus		*p*-Value
		No	Yes	
	*n* = 2041	*n* = 1465	*n* = 576	
Age (y)	31.6 ± 5.0	31.2 ± 4.8	32.5 ± 5.4	**<0.001**
Height (cm)	166 ± 7	166 ± 7	165 ± 8	**0.004**
Pre-pregnancy Weight (kg)	75.9 ± 17.5	74.0 ± 17.0	81.0 ± 17.8	**<0.001**
Pre-pregnancy BMI (kg/m^2^)	27.6 ± 7.8	26.7 ± 5.8	29.8 ± 11.1	**<0.001**
Nulliparous (%)	49.8	51.1	46.4	0.052
** *Ethnicity (%)* **				**0.044**
Northern European	79.6	80.9	76.3	
Moroccan	3.1	2.5	4.6	
Middle Eastern	1.4	1.5	1.3	
South Asian	2.4	2.3	2.8	
Mediterranean	2.4	2.7	1.8	
Indian descent/Surinamese	1.6	1.3	2.4	
Afro-Caribbean	0.7	0.8	0.6	
African	1.3	1.4	0.9	
Other	7.4	6.5	9.4	
Gestational age OGTT (weeks^+days^)	25^+4^ ± 1^+4^	25^+4^ ± 1^+4^	25^+4^ ± 1^+4^	0.588
BMI (kg/m^2^)	30.0 ± 9.4	29.3 ± 10.1	31.9 ± 6.9	**<0.001**
Systolic Blood pressure (mmHg)	111 ± 10	110 ± 10	114 ± 10	**<0.001**
Diastolic Blood pressure (mmHg)	63 ± 6	62 ± 6	64 ± 7	**<0.001**
Mean Arterial Pressure (mmHg)	79 ± 7	78 ± 7	81 ± 7	**<0.001**
BP ≥130 and/or ≥80 (%)	5.3	3.9	8.9	**<0.001**
Use of antihypertensive drugs (%)	13.5	14.1	12.0	0.201
** *Biochemistry* **				
HbA1c (%)	4.80 [4.60;5.0]	4.80 [4.60;5.0]	5.00 [4.80;5.20]	**0.019**
HbA1c (mmol/mol)	29 [27;31]	29 [27;31]	31 [29;33]	**0.019**
Cholesterol (mmol/L)	6.00 [5.30;6.80]	6.10 [5.40;6.80]	5.90 [5.20;6.60]	**<0.001**
HDL (mmol/L)	2.00 ± 0.48	2.05 ± 0.48	1.86 ± 0.45	**<0.001**
LDL (mmol/L)	3.23 ± 1.00	3.27 ± 1.00	3.14 ± 0.98	**0.006**
Triglycerides (mmol/L)	1.97 [1.54;2.47]	1.90 [1.49;2.36]	2.18 [1.75;2.70]	0.902
Protein to creatinine ratio	10.0 [8.28;12.60]	9.90 [8.10;12.40]	10.50 [8.60;13.30]	0.103
** *OGTT* **				
Fasting glucose (mmol/L)	4.9 ± 0.5	4.7 ± 0.3	5.3 ± 0.5	**<0.001**
Glucose load 1 h (mmol/L)	7.7 ± 1.9	7.0 ± 1.4	9.4 ± 1.8	**<0.001**
Glucose load 2 h (mmol/L)	6.5 ± 1.5	6.0 ± 1.2	7.8 ± 1.6	**<0.001**
** *Constituents of MetS* **				
Hypertension (%)	17.5	17.0	19.0	0.295
Obesity (%)	30.4	24.5	45.4	**<0.001**
Overweight (BMI > 25) (%)	60.9	54.7	76.6	**<0.001**
Dyslipidemia (%)	5.7	4.6	8.7	**<0.001**
Proteinuria (%)	1.3	1.0	2.0	0.104
** *Pregnancy outcomes* **				
GH (%)	11.2	10.4	13.0	0.096
PE (%)	4.8	4.6	5.2	0.544
HELLP syndrome (%)	0.7	0.7	0.9	0.659
Eclampsia (%)	0.0	0.0	0.0	NA
Intra uterine fetal demise (%)	0.2	0.3	0.2	0.683
GA delivery (weeks^+days^)	38^+5^ ± 2^+0^	38^+6^ ± 2^+0^	38^+3^ ± 1^+6^	**<0.001**
<37 weeks (%)	8.3	7.7	9.7	0.138
<34 weeks (%)	2.7	2.7	2.8	0.885
Caesarean section (%)	27.9	27.2	29.6	0.269
Instrumental delivery (%)	12.4	12.5	12.2	0.855
Shoulder dystocia (%)	1.8	1.6	2.3	0.299
Manual placental removal (%)	3.8	4.1	3.1	0.321
Blood loss (mL)	300 [200;500]	300 [200;500]	300 [200;500]	0.225
Blood loss >1000 mL (%)	10.4	10.5	10.1	0.803
** *Neonatal* **				
Boy (%)	54.0	55.4	50.4	**0.043**
Girl (%)	46.0	44.6	49.6	**0.043**
Birth weight (g)	3293 ± 873	3292 ± 590	3295 ± 1349	0.944
Percentile <10 (%)	9.8	10.4	8.2	0.121
Percentile >90 (%)	10.3	10.1	10.8	0.649
Congenital abnormalities (%)	3.8	3.7	4.2	0.600

OGTT: Oral glucose tolerance test; BMI: body mass index; BP: blood pressure; HbA1c: glycated hemoglobin; HDL: high-density lipoprotein; LDL: low-density lipoprotein; GH: gestational hypertension; HELLP: hemolysis elevated liver enzymes low platelets.

**Table 2 nutrients-14-02933-t002:** Effect of the number of constituents of the metabolic syndrome in absence or presence of GDM on large or small for gestational age infancy, preeclampsia and preterm birth.

Constituents	GDM-			p Trend	GDM+			p Trend
	0	1	≥2		0	1	≥2	
**LGA**	88/861 10.2%	47/465 10.1%	9/103 8.7%	0.714	32/235 13.6%	20/232 8.6%	9/88 10.2%	0.195
**SGA**	87/861 10.1%	53/465 11.4%	12/103 11.7%	0.441	19/235 8.1%	19/232 8.2%	7/88 8.0%	0.986
**PE**	21/861 2.4%	33/465 7.1%	11/103 10.7%	**<0.001**	6/236 2.5%	13/232 5.6%	11/88 12.5%	**<0.001**
**<37 weeks**	51/861 5.9%	45/465 9.7%	12/103 11.7%	**0.004**	19/236 8.1%	21/232 9.1%	15/88 17.0%	**0.035**
**<34 weeks**	17/861 2.0%	13/465 2.8%	7/103 6.8%	**0.011**	5/236 2.1%	5/232 2.2%	6/88 6.8%	0.062

GDM: Gestational diabetes mellitus; LGA: large for gestational age; SGA: small for gestational age; PE: preeclampsia.

**Table 3 nutrients-14-02933-t003:** Adjusted odds ratios of LGA, SGA, PE, <37 wk and <34 wk gestational age at birth of the different present conditions at OGTT (GDM, hypertension, obesity, dyslipidemia, and proteinuria, and having ≥2 constituents of the metabolic syndrome. Adjustments were made for the other constituents of the metabolic syndrome, age, Northern European ethnicity and gestational age at time of OGTT.

OR (95%CI)		LGA	SGA	PE	<37 Wk	<34 Wk
** *Adjusted OR (95%CI)* **		10.3%	9.8%	4.8%	8.3%	2.7%
**GDM**	28.2%	1.08	0.76	1.15	1.29	1.05
		(0.79–1.47)	(0.54–1.08)	(0.74–1.78)	(0.92–1.80)	(0.58–1.89)
*aOR*		*1.02*	*0.78*	*1.18*	*1.45*	*1.23*
		(*0.72–1.45*)	(*0.54–1.12*)	(*0.70–1.97*)	(*0.99–2.12*)	(*0.64–2.34*)
**Hypertension**	17.5%	**0.50**	**1.51**	**4.93**	**2.42**	**2.96**
		(0.31–0.79)	(1.07–2.14)	(3.25–7.48)	(1.71–3.42)	(1.88–5.68)
*aOR*		** *0.46* **	** *1.59* **	** *3.14* **	** *2.17* **	** *2.96* **
		(*0.28–0.77*)	(*1.09–2.31*)	(*1.94–5.10*)	(*1.46–3.23*)	(*1.59–5.53*)
**Obesity**	30.3%	1.19	0.76	0.93	0.97	0.94
		(0.88–1.61)	(0.55–1.07)	(0.59–1.45)	(0.69–1.37)	(0.52–1.69)
*aOR*		*1.17*	*0.81*	*0.79*	*0.99*	*1.02*
		(*0.84–1.63*)	(*0.57–1.15*)	(*0.47–1.31*)	(*0.68—1.44*)	(*0.55–1.91*)
**Dyslipidemia**	5.7%	1.30	0.66	**2.50**	**1.98**	2.07
		(0.74–2.28)	(0.32–1.38)	(1.33–4.73)	(1.15–3.40)	(0.87–4.93)
*aOR*		*1.15*	*0.71*	** *2.35* **	*1.69*	*1.90*
		(*0.61–2.16*)	(*0.34–1.50*)	(*1.09–4.73*)	(*0.92–3.11*)	(*0.76–4.76*)
**Proteinuria**	1.3%	0.34	**2.79**	**13.68**	**7.36**	**9.52**
		(0.05–2.55)	(1.11–7.03)	(6.03–31.03)	(3.29–16.50)	(3.45–26.32)
*aOR*		*NA*	*1.84*	** *12.64* **	** *4.72* **	** *5.16* **
			(*0.60–5.70*)	(*4.37–36.54*)	(*1.67–13.35*)	(*1.33–20.08*)
**≥2 constituents**	9.4%	0.89	1.00	**3.07**	**2.01**	**3.20**
		(0.54–1.49)	(0.61–1.65)	(1.86–5.07)	(1.29–3.13)	(1.68–6.10)

OR: Odds ratio; LGA: large for gestational age; SGA: small for gestational age; PE: preeclampsia; GDM: gestational diabetes mellitus.

## Data Availability

The data presented in this study are available on request from the corresponding author.

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
