# Peer review of "Second-Trimester Constituents of the Metabolic Syndrome and Pregnancy Outcome: An Observational Cohort Study"

_nutrients, 2022, doi:10.3390/nu14142933_

Round 1

Reviewer 1 Report

The paper has the merrit to investigate the metabolic syndrome related to GD. Thde conclusion of the authors is that if the diabetes is well control the perinatal outcome is good. The paper respects the scientific rigour. 

I think the article bring the evidence that an early diagnosis is the key of implemting a correct therapy to prevent complications.

Reviewer 2 Report

Major revision

The manuscript: “Second Trimester Constituents of the Metabolic Syndrome and Pregnancy Outcome; an observational cohort study” studied the relationship between metabolic syndrome and impaired gestational glucose tolerance on subsequent maternal and neonatal outcome. (preeclampsia, altered fetal growth and preterm birth).

The manuscript is well written. The conclusion are in concordance with the results.

Why the obesity are not in relation with neonatal outcome?

Minor revision

Please delete the point after the title.

On page 1 line 26 what is OGTT? Please explain the acronym.

On page 1 line 30: ….. the risk of SGA (aOR 1.59), PE (aOR 3.14), preterm birth <37 weeks (aOR 2.17) and <34 weeks 30 (aOR 2.96). Please delete “a” from parentheses.

In tables acronyms are not explained.
